# Case Study on Increasing Breeding Value Estimation Reliability of Reproductive Traits in Serbian Highly Prolific Large White and Landrace Sows

**DOI:** 10.3390/ani12192688

**Published:** 2022-10-06

**Authors:** Nenad Stojiljković, Dragan Radojković, Zoran Luković, Marija Gogić, Čedomir Radović, Mladen Popovac, Dubravko Škorput

**Affiliations:** 1Department of Pig Breeding and Genetics, Institute for Animal Husbandry, Highway 16, 11080 Belgrade, Serbia; 2Department of Animal Science, Faculty of Agriculture, University of Belgrade, Nemanjina 6, 11080 Belgrade, Serbia; 3Department of Animal Science and Technology, Faculty of Agriculture, University of Zagreb, Svetošimunska cesta 25, 10000 Zagreb, Croatia

**Keywords:** pigs, litter size, connectedness rating, reliability, estimated breeding value

## Abstract

**Simple Summary:**

In the Republic of Serbia, pig selection in recent decades has been based on genetic improvement of growth and carcass quality traits. Genetic improvement of reproductive traits of pigs was based on the so-called phenotypic selection. The introduction of modern information systems and the availability larger dataset have opened the possibility to perform genetic estimation of reproductive traits within the main breeding programme of the Republic of Serbia. Using the methods of gene flow and connectedness evaluation, our study investigated the possibility of improving the reliability of estimating the breeding value of reproductive traits in highly productive sows. We believe that these methods could lead to a systematic improvement of the genetic value of reproductive traits in sows. Thus far, none of the methods for estimating the degree of connectedness between herds in pigs has been used in the preparation of the National Breeding Programme of the Republic of Serbia.

**Abstract:**

This study investigated the influence of the degree of connectedness on the reliability of the estimated breeding values (EBVs). The focal trait in the study was the number of piglets born alive (NBA) from sows of the highly prolific Large White and Landrace sows. An analysis included total of 58,043 farrowing’s during the 2008–2020 period. BLUP procedure was used to estimate the breeding values for NBA for the three herds separately and after merging all three herds into one herd. The model for EBV estimation included the following fixed factors: parity, genotype, seasons, litter sire, herds, sow age at farrowing, weaning-conception interval, length of previous lactation, and the following random effects: common litter environment, permanent litter environment, and direct additive genetic effect of animal. Heritability values for NBA ranged from 0.048 to 0.097, depending on the data included in the analysis. The connectedness between herds was analysed using the connectedness rating (CR) and the gene flow (GF) methods. CR among the observed herds ranged from 0.245 to 0.994%, depending on the data included. The exchange of genetic material between all three herds was determined using GF method. The high degree of connectedness determined by the CR and GF method had a strong effect on EBV reliability. The average EBV reliability ranged from 0.520 to 0.867, depending on the data included. The increase in average reliability was observed in both cases when the data were added, both in the analysis of average reliability for purebred animals and when crossbreeds were added, where an increase in this value was also observed. The increase in average EBV reliability is a consequence of the greater amount of information included in the joint evaluation. In conclusion, we believe that our research will improve EBV reliability and help in further selection work in the Republic of Serbia.

## 1. Introduction

Litter size traits are relatively easy to measure and monitor under production conditions and are, therefore, suitable for inclusion in selection programmes. Low heritability, negative correlation between direct additive genetic and maternal influence, expression only in females, and a relatively late age at which these traits are expressed in animals are the main reasons for slow improvement of litter size traits in the past. On the other hand, the presence of a relatively large direct additive genetic variance for litter size and the availability of data on numerous relatives allow successful selection for litter size. An important step in this direction is the use of the Linear Mixed Model (LMM) to estimate breeding value in domestic animals. The LMM method is now a standard procedure in pig production to evaluate the parameters of phenotypic and genotypic variability using the Restricted Maximum Likelihood (REML) method and to EBV of animals based on Best Linear Unbiased Prediction (BLUP) [1,2].

Numerous studies have confirmed that genetic progress in the overall reproductive performance of sows can be most effectively achieved by selection for litter size [3,4,5,6,7,8,9,10,11]. Postnatal litter size can be described as the NBA, stillborn, total born and weaned piglets. The number of weaned piglets is even more important economically than the size of the litter at birth. However, due to the widespread technological procedure of standardisation of litters of different farrowing at about the same time this trait is of secondary importance, right after the NBA. A high genetic correlation between the total number of piglets born and the NBA means that selecting for only one of the two traits is sufficient. Selection for the total number of piglets born also increases the number of stillborn piglets. Therefore, the NBA is a trait that is the most common choice for improving litter size in most breeding and selection programmes [4,5,12].

The estimation of breeding values using BLUP method in countries with developed and modern pig production is today a routine method for selecting the best individuals. This method makes it possible to estimate the breeding value of animals and their ranking within and between herds. The reliability of the breeding value estimation of animals from different herds depends on the degree of connectedness between them. The higher the connectedness, the more reliable the comparison of individuals from different herds seems to be. Higher genetic connectedness of the herd results in both more reliable EBV and reduced bias. If herds are insufficiently connected the comparison of breeding values between animals from different herds may be biased. Therefore, measuring the across-herd connectedness is important in order to achieve a more reliable genetic evaluation among pig herds (between a large number of farms/herds) [13,14,15,16,17,18,19].

The effect of connectedness in terms of differences between EBVs between and within herds depends on trait heritability [13]. A heritability for a back fat thickness ranges from 20% to 40%, while heritability for the NBA varies from 5% to 15%. These differences in the heritability of certain traits can have an impact on the differentiation of breeding values across herds. By examining the benefits of genetic connectedness between pig herds, the results show that average inbreeding coefficients decreased while cumulative selection responses increased [20]. The selection response is influenced by both genetic connectedness and trait heritability. For traits with low heritability, higher genetic connectedness between herds enables faster selection results. In many countries, commercial breeding programs place great emphasis on improving low-hereditary traits in sows. As already mentioned, the BLUP method enables the estimation of the breeding value of animals between and within the herd, with a greater effect of selection being achieved if the herds are connected at the genetic level.

The aim of measuring the connectedness between herds is to obtain an indication of accuracy and to reduce bias when estimating breeding value in different herds. Several methods have been developed for this purpose. One of the most adequate methods of measuring connectedness is the prediction error variance (PEV) of the difference in EBVs between individuals in different herds [21]. As this method of measuring connectedness/association requires complex computational operations to obtain variance in predicting differences in EBVs, the authors of [22] proposed three simpler methods: genetic drift variance, gene flow (GF), and effect estimation variance. The latter has the highest correlations with PEV differences in the estimation of breeding value. As the calculation of all differences in the pair between animals, for each possible pair in the herd, is computationally demanding, the connectedness rating (CR) method has been developed [13,22].

CR is expressed as a correlation between the assessments of herd effects, and it is strongly associated with PEV [13,22]. In addition, CR is less dependent on other methods for size and variation between herds. CR is a statistical method based on measuring the accuracy of comparing EBVs, not genetic links. Unrelated animals may be connected if tested on the same farm or within the same management group. A method such as “gene flow”, which is based on the degree of genetic links between animals, and not on statistical measurement of connectedness, may often give inferior results compared to CR [13,18].

The CR method has been routinely used in national genetic evaluation in Canada, Australia, and many other countries. The advantages of the practical application of the CR method are multiple. They are primarily reflected in the simplicity of calculation [16]. The CR method enables breeders to access a broader genetic base on which the selection will be carried out, and thus, increase the intensity of selection. In the Republic of Serbia, no method of assessing the connectedness between herds has been used so far in the development of the national breeding program. Therefore, the across-herd connectedness will be analysed in this paper due to its impact on the reliability of the estimate of breeding value and consequently on the effect of selection. The objectives of this study are to determine the degree of across-herd connectedness using the CR method and to determine the reliability of breeding values estimated by BLUP method both within a single herd and upon merging multiple herds into one.

According to a previous practice in the Republic of Serbia, the estimation of breeding value of reproductive traits in sows has been conducted individually per herds. By introducing a unique information system, the possibility of simultaneous genetic evaluation of larger number of herds has been opened. The results obtained in this research may indicate certain steps to be taken in order to increase reliability of estimating breeding value in highly prolific sows in the Republic of Serbia. The model applied and genetic parameters estimated could be used during a practical estimation of breeding value in sows.

## 2. Materials and Methods

### 2.1. Phenotypes and Genotypes

Data on litter size and pedigree data were provided by the Department of Animal Husbandry, Faculty of Agriculture, University of Novi Sad. The dataset included three farms (A, B, and C) over a period of 14 years (2007–2020). The reproductive traits of the four most common breeds on farms in the Republic of Serbia (Landrace-L, Large White-LW and the crosses LxLW and LWxL) were analysed. The complete dataset (herd-ABC) contained records of 58,043 farrowings (Table 1). The NBA was a focal of this study.

The pedigree file required for this analysis was created for at least three generations of ancestors. The complete pedigree file (Table 1) for all three farms analysed contained 23,453 individual animals of which 18,708 were individual animals with production data. The percentage of animals without data for both parents (so-called base animals) for Herd-ABC was 13.02% (Table 1). All sows used in the analysis are registered in the Herd Book managed by the Department of Animal Science at the Faculty of Agriculture in Novi Sad. Farm B has its own AI centre. Pedigree records were analysed using the CFC software package v.1.0, Sargolzei et al., Nigata, Japan [23], while descriptive statistics was calculated using software package SAS Inst. Inc., Cary, NC, USA [24].

### 2.2. Statistical Analyses

#### 2.2.1. Applied Model

In order to determine the significance of the fixed effects on the dependent variable, data processing was performed using the GLM procedure of the software package AS Inst., Inc., Cary, NC, USA [24]. The systematic effects included in the model for the assessment of genetic parameters were selected through a “step by step” procedure according to the criterion of statistical significance of the expressed influences. Data were analysed using the following linear mixed model:(1)$$\left[y_{\mathrm{ijklmnop}}\right]\;=F_{i}+S_{j}+G_{k}+B_{l}+H_{m}+b_{1i}(x_{\mathrm{ijklmnopq}}-\overset{-}{x})+b_{2i}(x_{\mathrm{ijklmnopq}}-\overset{-}{x}{)}^{2}+\left[Z_{n}+\;b_{3}(z_{\mathrm{ijklmnopq}}-z^{¯})\right]+l_{o}+p_{\mathrm{ijklmnopq}}+a_{\mathrm{ijklmnopq}}+e_{\mathrm{ijklmnopq}}$$
where y_fijklmnopq_—a manifestation of the observed trait of the individual animal in sows’ litters, F_i_—fixed effect of parity, S_j_—fixed effect of the season of successful mating, presented as combination of the year and month, G_k_—fixed effect of the female’s genotype B_l_—fixed effect of the litter sire, H_m_—fixed effect of the herd (used when all three herds were analysed together), b_1i_(x_ijklmnopq_—
$\overset{-}{x}$) and b_2i_(x_ijklmnopq_—$\overset{-}{x}$)^2^—linear and square regression effect, respectively, effect of the sow age at farrowing (x) nested within the parity, $Z_{n}$—fixed effect of the class of weaning to conception interval, $b_{3}\left(z_{\mathrm{ijklmnopq}}-z^{¯}\right)$—linear effect of the duration of previous lactation, l_o_—random effect of the common litter environment, p_ijklmnopq_—random effect of the permanent environment, a_ijklmnopq_—direct additive genetic effect of animal, i.e., breeding value, and e_ijklmnopq_—residual.

The model used to estimate the components of variance and breeding value for the NBA can be presented in matrix form as follows:

y = Xb + Z_l_l + Z_p_p + Z_a_a + e
(2)

where y—an observation vector for the analysed trait (NBA), b—the vector of unknown parameters for the fixed part of the model (season—defined as the interaction of year and month of successful mating, genotype, litter sire, parity, age at farrowing, duration of previous lactation, and duration of weaning to conception interval); l, p, and a—vector of unknown parameters for the effect of the common environment of the litter in which the sows were born (reared), permanent influence of the environment in the litters of sows and direct additive genetic effect of the animal or breeding value, e—vector of random residues, and X, Z_l,_ Z_p_, and Z_a_—event matrices that connect phenotype records with corresponding effects.

The assumptions are as follows:$$E\left(y\right)=\;\mathrm{Xb},\;l\sim \;N\left(0,{\;I\sigma }_{l}^{2}\right),\;p\sim N\left(0,{\;I\sigma }_{p}^{2}\right),\;a\sim N\left(0,{\;A}_{a}^{2}\right),\;e\sim N\left(0,{I\sigma }_{e}^{2}\right)$$
where A is the relation matrix counter.
$$E\;\left[\begin{array}{c} \begin{array}{c} l\\ p\\ a \end{array}\\ e \end{array}\right]=\left[\begin{array}{c} 0\\ \begin{array}{c} 0\\ 0\\ 0 \end{array} \end{array}\right],\;\mathrm{var}\;\left[\begin{array}{c} l\\ \begin{array}{c} p\\ a\\ e \end{array} \end{array}\right]=\left[\begin{array}{cccc} I_{l}\sigma_{l}^{2} & 0 & 0 & 0\\ 0 & I_{P}\sigma_{p}^{2} & 0 & 0\\ 0 & 0 & {A\sigma }_{a}^{2} & 0\\ 0 & 0 & 0 & I_{e}\sigma_{e}^{2} \end{array}\right]$$

#### 2.2.2. Gene Flow

The extent and direction of gene flow between herds was estimated using the gene flow (GF) method [21]. The connectedness between herds by GF method is calculated by multiplying the matrix XZTQ, where Q denotes the base animals with respect to their herd and T is a lower triangular matrix that tracks the gene flow from one generation to the next, where the linkage counter is the matrix A = T’WT and W the diagonal of the matrix of Mendelian sampling variants. Gene flow between herds was calculated for a period of 13 years. One dataset contained data for pure breeds and the other data on all genotypes in the herds (L, LW, LxLW, and LWxL). Matrix formation and GF calculation was done with the RStudio software package [25], using the following software packages: pedigreemm [26], pedigree [27], and MatrixModels [28].

#### 2.2.3. Connectedness Rating

Connectedness rating measure (CR) was used to assess the association between herds [22]. The CR calculation program used for this method was downloaded from the Canadian Pig Improvement Centre website [29]. CR is defined as the correlation between the assessment of herd effects:$$\mathrm{Cr}=\frac{\mathrm{Cov}\left({\hat{h}}_{i},{\hat{h}}_{j}\right)}{\sqrt{\mathrm{Var}\left({\hat{h}}_{i}\right)\mathrm{Var}\left({\hat{h}}_{j}\right)}}$$
where variances Var(${\hat{h}}_{i})$ and Var (${\hat{h}}_{j})$ and covariances Cov$\left({\hat{h}}_{i},{\hat{h}}_{j}\right)$ for the corresponding herd effects were estimated by directly solving the left-hand side of the mixed model equation (MME). The MMEs for Model 1 are as follows:$$[\begin{array}{cccc} X^{T}X & X^{T}Z_{l} & X^{T}Z_{d} & X^{T}Z_{a}\\ Z_{l}^{T}X & Z_{l}^{T}Z_{l}+{I\sigma }_{e}^{2}\sigma_{a}^{-2} & Z_{p}^{T}Z_{l} & Z_{l}^{T}Z_{a}\\ Z_{p}^{T}X & Z_{p}^{T}Z_{l} & Z_{p}^{T}Z_{p}+{I\sigma }_{e}^{2}\sigma_{a}^{-2} & Z_{p}^{T}Z_{a}\\ Z_{a}^{T}X & Z_{a}^{T}Z_{l} & Z_{a}^{T}Z_{d} & Z_{a}^{T}Z_{a}+A^{-1}\sigma_{e}^{2}\sigma_{a}^{-2} \end{array}]\;[\begin{array}{c} \begin{array}{c} b\\ l\\ p \end{array}\\ a \end{array}]\;=\;\left[\begin{array}{c} \begin{array}{c} X^{T}y\\ Z_{l}^{,}y\\ Z_{p}^{,}y \end{array}\\ Z_{a}^{,}y \end{array}\right]$$

The number of farms, individuals, and fertility data used to calculate CR are shown in Table 1. CR was calculated for different combinations of data. In the first case, the CR between the herds was calculated using data on purebred individuals separately for Landrace and Large White. The CR between the herds was then calculated using a common set of Landrace and Large White data. Finally, CR was calculated for both pure breeds and crossbreeds (LxLW, LWxL).

#### 2.2.4. Reliability of Estimated Breeding Values

Reliability of EBV (r^2^) was calculated as follows:

r^2^ = 1 − (PEV/VarA)

where PEV is the prediction error variance and VarA is the additive genetic variance in the analysed population [1]. PEV was calculated by solving the left side of the MME equation directly.

To estimate breeding values and their reliability, the variance components were assessed. The same model (1) was used for the estimation of variance components, breeding values, and their reliability in order to allow a proper comparison of the breeding advantages between the farms. The components of variance were first assessed for each farm separately and then together for all three farms. VCE 6 [30] software was used to evaluate the variance components. In this study, when determining variance components and estimating breeding value by means of a BLUP animal model, the analysed traits have been treated as the traits occurring repeatedly during a production lifespan repeatability treatment. Reliability was calculated for all EBVs and average reliability was delivered for each individual farm and for all three farms together. Estimation of breeding value was performed using the PEST program [31].

## 3. Results

Variance components were estimated both for each individual herd and after merging all three herds into one dataset. First, variance components were estimated for the datasets containing the pure breeds Large White and Landrace (Table 2). All estimated variance components, except for the effect of shared litter environment, were highest in herd C. Direct additive genetic variance was lowest in herd A. Heritability estimates ranged from 0.048 in herd A to 0.095 in herd C. The random litter effect related to the litter in which the animals were born/raised was the lowest of all random effects, ranging from 0.000 to 0.028. A permanent effect of a breeding animal, caused by the successive litters of the animal and presented in terms of phenotypic variance, explained 0.036 to 0.093 variability.

Upon adding crossbreed data to the purebred dataset, the variance components for each farm separately and upon merging data into one dataset have been estimated. Table 3 shows no variance components for herd B because this herd contained did not contain crossbreed data. As in the analysis of pure breeds, the highest values of the estimated variance components were obtained for farm C, except for a random effect of the litter in which the animals were born. The direct additive genetic variances were quite similar for all three analysed datasets, ranging from 0.912 to 0.790. A slight increase in additive genetic variance was observed when all three herds were combined into one. The highest value for heritability was observed in herd C. The permanent effect of breeding animals produced by their successive litters ranged from 0.064 to 0.089. For the random effect of the litter in which animals were born/bred, relatively low values between 0.009 and 0.017 were obtained.

Measuring the degree of across-herd connectedness is very important for efficient estimation of breeding value between herds. The connectedness between herds (A, B, and C) for Landrace and Large White is shown in Table 4. The values of CR for both breeds were well above the minimum value of 1.5% for the NBA, which is required for genetic evaluation between herds [13].

The level of connectedness between the observed herds was quite high for the Large White breed, ranging from 0.864% to 0.983%. The highest CR value for the Large White breed was 0.983% between herds B and C. The correlation between herds A and C was slightly lower at 0.864 %. A high CR between B and C herds was also found for the Landrace breed (0.895%), while the correlation between A and B flocks or A and C herds the connectedness was lower, ranging from 0.245% to 0.274%.

The connectedness upon merging into one database for pure Landrace and Large White as well as the pure breeds and their crossbreeds is shown in Table 5. CR for pure breeds between all herds was high, ranging from 0.828% to 0.987%. The highest CR value for the purebred dataset was recorded between herds B and C (0.987%). After adding F1 crosses to the pure breeds, there was an increase in CR values between A–B and A–C herds. The connectedness between B and C herds had a slightly lower value of 0.980% compared to the pure breed dataset of 0.987%.

Gene flow, i.e., the genetic contribution of animals from one herd to another, is shown in Figure 1 and Figure 2. This method expresses the genetic contribution of animals from a given herd to another as the proportion of genes in a herd that originate from other herds. The GF for the herds studied was determined for all three herds in both directions. Gene flow for pure breeds was thoroughly monitored both before and after crossbreeds were included in the dataset. The exchange of genetic material was determined between all three herds. Only in the direction of herd C, from herd A, there was no exchange of genetic material, when only pure breed was observed.

After the addition of crossbreeds, the largest GF observed in the period from 2008 to 2020 was between B and C herds. The share of genes from herd B found in herd C was 0.74%. A large percentage of genes derived from farm B was also found on farm A (0.65%). The lowest number of exchanged animals was between herds A and C (0.09 and 0.02%). The percentage of genes originating from other farms on farm B ranged from 0.02% to 0.10%. Herd C had the lowest genetic contribution to the other herds.

Increasing the accuracy of EBV for reproductive traits in sows is important because traits from this group have low heritability and manifest relatively late in females. The reliability of breeding value estimation for sow reproductive traits depends on the amount of information used in the estimation, in addition to the indicated level of connectedness between herds. The average reliability of the EBV when using purebred data is shown in Table 6. Herd A had the lowest reliability of the EBVs with an average value of 0.520. For the analysed herds, the reliability ranged from 0.519 to 0.867. Herd C had the highest average reliability of the EBVs of 0.867.

Comparison of the reliability of breeding value obtained by BLUP method for datasets containing pure breeds and crossbreeds is shown in Table 7. After adding crossbreeds to datasets with purebreds, the average reliability of EBVs in herds A and ABC increased. Herd C had slightly lower values compared to the pure breed dataset. Generally, reliability has increased with the addition of more information in estimating breeding value. The average reliability ranged from 0.812–0.846. The reliability of EBV in Table 7 for herd B was not shown because it contained no crossbreed data.

## 4. Discussion

The NBA in litter is affected by a number of factors which must be included into models in order to estimate variance genetic component as accurately as possible. Of all the random effects identified, the additive genetic variance had the highest proportion of total phenotypic variance. The values obtained for the additive genetic variances (0.508 to 0.970) indicate that it is possible to perform successful selection in the studied population based on the NBA. A random effect of the litter in which the sows were born/raised and a permanent environmental effect both contributed to a more accurate estimate of the variance component. The proportion of common litter environment variance being 0.181 in total phenotypic variance of NBA is deemed significant since only 10.634 of total variability can be explained by random effects included into the model in case when all the three farms were simultaneously analysed.

Heritability values for NBA ranged from 0.048 to 0.095 when purebred data were used, and when crossbreeds were added, values ranged from 0.077 to 0.097. These values are similar to the results found in many research studies [5,32,33]. Heritability values increased when all three farms were analysed simultaneously, which may be associated with the presence of genetic links between farms and a greater amount of data included in the analysis.

The presence of some degree of connectedness between herds is a basic requirement for estimating the breeding value with satisfactory accuracy and reliably comparing animals between herds [14,18]. The absence of genetic links between herds can have a negative impact on EBV bias. To highlight the importance of the degree of connectedness, many researchers have investigated the influence of genetic association on EBV [14,15,16,17,18,19,34]. However, there is little research indicating the importance of connectedness between herds when estimating breeding values of litter size traits in sows with low heritability.

In the results presented, there is a high degree of connectedness between all the herds included in the analysis. This high degree of connectedness is explained by the fact that herd B has an AI centre which provides boar semen to the other two herds. There is also trade in breeding animals between the herds. Such a high connectedness between herds indicates that EBVs can be compared with high reliability between herds, which is also shown in the following studies [13,14,15,19].

The first objective of this study was to determine the connectedness between herds of highly prolific sows and to determine its significance. The CR method of determining connectedness between herds was chosen because studies by many authors have shown that indicating connectedness using this method is less computationally demanding [13,15,16]. Lower computational requirements when using the CR method facilitate its practical application in genetic evaluation between herds. Based on the high degree of connectedness, it is possible to recommend an estimation of the breeding value between the analysed herds. Although the values of CR compared to Large White were lower between herds when only the Landrace breed was analysed, they were much higher than the 1.5% recommended for reproductive traits [13], indicating that genetic evaluation between herds can be performed with high accuracy.

After merging data from pure breeds and crossbreeds into one dataset and analysing the data, an increase in CR was observed between all analysed herds, especially after the addition of crossbreeds, which was not the case in a previous study [18]. This shows that it is important to include more data in the determination of the association and the subsequent estimation of breeding values, which is in line with the results presented in [15]. Higher values of CR, after the addition of crossbreeds, in addition to a larger number of data, are also associated with the fact that boars from the AI centre of one genotype were used to inseminate seedlings of another. The mating of animals of different genotypes resulted in an increase in CR. The high connectedness in this study after the addition of crossbreeds indicates that the bias in EBV can be reduced [13].

In order to determine the direction of gene flow between herds the GF method was used [18,21,35]. This method provides additional information on inter-herd connectedness. GF monitors gene transfer from one herd to another by forming groups of animals based on the origin of the herd. These groups should not be confused with genetic groups that are formed in order to represent different genetic groups in the population [21].

This method is suitable when the only purpose is to determine whether there was gene flow between herds. When data from three farms were included in the analysis of genetic variability in a dataset, the model determinant weighed zero and the programme found the dependence of the data in the inverse matrix, so that the parameters could not be calculated because there were no genetic links between the herds [32]. In such cases, GF is suitable for a quick and easy calculation and for determining the existence of genetic links between the herds.

The highest GF in the herds studied was from herd B towards herds A and C in both sets of data, which is consistent with the results obtained using the CR method. Additionally, the results of the CR method showed the greatest connectedness between these herds. This is explained by the fact that herd B has an already-mentioned AI centre and distributes boar semen to the other herds. The only case where there was no exchange of genetic material was from herd C to herd A for a pure breed dataset. In the opposite direction, a small percentage of genes were found in herd A. Among these herds, there was no exchange of genes for the reason that they are commercial-type farms that primarily produce high-yielding gilts of F1 generation and piglets for fattening. When crossbreeds were added to the dataset for analysis, there was a flow of genes from herd C towards herd A.

When selecting animals raised in different herds, the accuracy of the estimation of breeding value can be strongly influenced by the connectedness between the herds. If several herds are genetically evaluated together, the estimated EBVs may be overestimated. If the herds are well connected, the reliability of the EBVs can be expected to be higher in a joint evaluation, as information from one herd contributes to the other [18,19,36]. The high level of connectedness determined by the CR and GF method strongly influenced the average reliability of EBV, which is consistent with the results presented in previous studies [13,14,18,35].

Increasing the reliability of EBV is very important for the litter traits, because they manifest only in female individuals and have low heritability. Publications [13,15,20] state that heritability and across-herd connectedness can influence the selection results. Low heritability of reproductive traits in sows and high across-herd connectedness had an impact on selection results. It is assumed that the selection response, i.e., the reliability of EBV, was influenced by the low heritability in this study, in addition to high connectedness. Since low-hereditary traits were analysed in this study and the effect of selection depends on its heritability coefficient, the aim is to increase the effect of selection along with the increased accuracy.

The results of this study show that the estimation of breeding values between herds that are genetically connected can strongly influence the genetic progress of the population under selection. If genetic association is not considered when estimating breeding values of animals from different herds, it is assumed that the average genetic values of the analysed traits of all the herds are the same. This assumption may have a negative impact on the estimation of breeding values [15].

In addition to across-herd connectedness, an important factor influencing the reliability of the EBVs was the increased number of individuals in the pedigree and phenotype records. Upon merging the herds into a single dataset, there occurred an increase in the average EBV reliability. In both cases, when the average reliability for the pure breed was analysed and after the addition of crossbreeds, this value increased. The increase in the average reliability of EBVs is probably a consequence of the increased volume of information included in the joint evaluation, as reported in the research of many authors [18,36,37,38]. The reliability of the EBVs for the NBA was much higher than in [39]. These authors reported much lower values for the reliability of EBVs in highly prolific sows, i.e., 0.33 for the NBA compared to 0.74 recorded in our study. Even though the data on pedigree and own phenotype were used in both studies, the sample was much smaller in the estimation of BV in the mentioned study [39], which can be related to the amount of information in the pedigree file and the smaller volume of data on phenotype.

Since the increased amount of data in the breeding value estimation has a positive effect on the reliability of EBV, further research of this type should be conducted on a larger number of farms. The inclusion of a larger number of farms would increase the population on which the selection is carried out. In this way, the response of the selection would be directly influenced, i.e., the NBA would increase. The inclusion of more farms would also allow smaller farms to compare their animals with larger farms [40,41]. The CR method allows breeders to access a broader genetic base, on which selection is carried out, thus increasing the intensity of selection.

## 5. Conclusions

The heritability values (0.048 to 0.097) obtained for the NBA indicate that successful selection is possible in the investigated pig population of the Republic of Serbia. The sufficient connectedness level and the direct genetic links allow the estimation of the breeding value with high reliability. High values of average reliability from 0.520 to 0.867 in this study confirm this. Reliability increased with the addition of pedigree information and phenotype data. This shows that an increase in EBV reliability is possible by including additional data. High values for EBV reliability are associated with the large amount of data and good connectedness between farms. High reliability has an important role, especially in the selection for traits with low heritability. The high values of CR between the analysed herds in this study suggest that the use of artificial insemination in the selected population may indirectly increase EBV reliability. In addition to the increased reliability of EBVs, high CR values indirectly affect the positive selection response. This is important because the effects of selection are directly related to the accuracy of breeding value estimation. It should be noted out that besides the lack of systematic selection an average NBA in the analysed population is not low. For this reason, we should be cautious when making recommendations regarding future pig breeding decisions. We believe that our study can contribute to a better understanding of breeding value estimates in highly productive sows in the Republic of Serbia. Furthermore, the results obtained could have an impact on the future direction of pig breeding.

## Figures and Tables

**Figure 1 animals-12-02688-f001:**
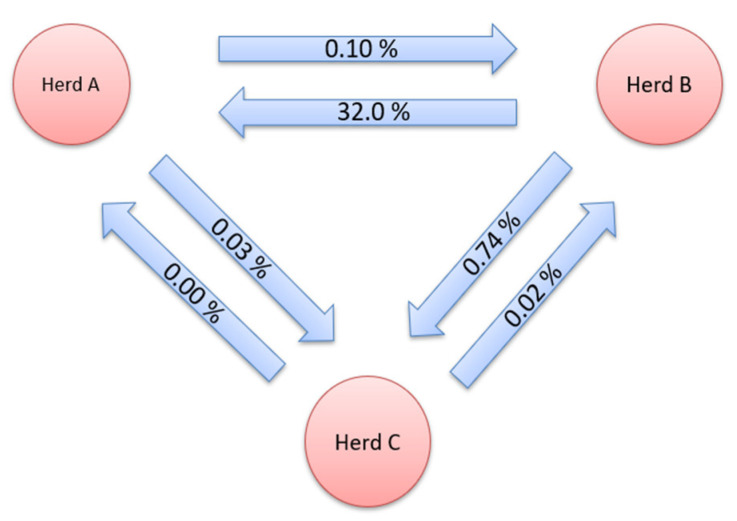
Gene flow between herds for purebred data.

**Figure 2 animals-12-02688-f002:**
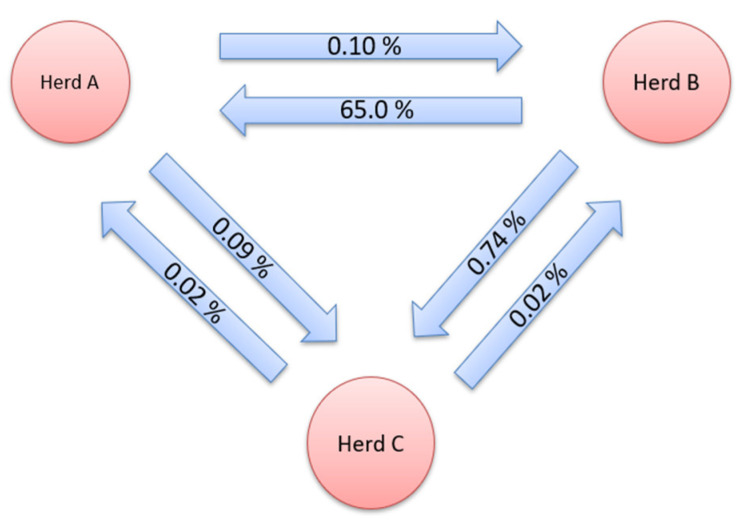
Gene flow between herds for purebred and crossbreed data.

**Table 1 animals-12-02688-t001:** Structure of data and pedigree files used in the analysis.

Herd Name	Phenotypes			Litter Genotype				Pedigree File		
	N	$\overset{-}{x}$-NBA	SD	L	LW	LXLW	LWXL	No. of Animals	% Base Animals	N2
A	36,200	16.18	3.58	591	1111	21,948	12,550	15,478	12.96	10,192
B	7823	14.00	3.79	3510	4313	/	/	4886	20.91	3160
C	14,020	16.88	3.84	988	853	5801	6378	10,878	18.46	5609
ABC	58,043	16.05	3.77	5098	6277	27,749	18,928	23,453	13.02	18,708

N—number of litters, $\overset{-}{x}$-NBA—average number of piglets born alive, SD—standard deviation, No. of animals—Number of animals in pedigree file, % Base animals—% base animals in the pedigree file, N2—number of individual animals with production results in the pedigree file.

**Table 2 animals-12-02688-t002:** Estimates of variance components and ratios with standard errors for NBA using purebred data.

Herd	Estimates of Variance Components for Purebred Data				
	Var(a)	Var(p)	Var(l)	Var(e)	Var(ph)
A	0.508 + 0.455	0.951 + 0.543	0.000 + 0.000	9.053 + 0.415	10.513
B	0.602 + 0.191	0.428 + 0.195	0.332 + 0.124	10.318 + 0.211	11.681
C	1.615 + 0.590	1.248 + 0.752	0.000 + 0.000	14.058 + 0.648	16.922
ABC	0.825 + 0.196	1.171 + 0.214	0.189 + 0.117	10.324 + 0.174	12.511
	**h^2^**	**p^2^**	**l^2^**	**e^2^**	
A	0.048 + 0.042	0.090 + 0.051	0.000 + 0.000	0.861 + 0.033	
B	0.051 + 0.016	0.036 + 0.016	0.028 + 0.010	0.883 + 0.014	
C	0.095 + 0.034	0.073 + 0.044	0.000 + 0.000	0.830 + 0.035	
ABC	0.065 + 0.015	0.093 + 0.017	0.015 + 0.009	0.825 + 0.012	

Var(a)—direct additive genetic variance; Var(l)—variance of common litter environmental effect; Var(p)—permanent environmental variance; Var(e)—residual error variance; Var(ph)—phenotypic variance; **h^2^**—direct heritability; **p^2^**—proportion of permanent environmental effect; **l^2^**—proportion of common litter environmental effect; **e^2^**—proportion of residual error variance.

**Table 3 animals-12-02688-t003:** Estimates of variance components and ratios with standard errors for NBA using purebred and crossbreed data.

Herd.	Estimates of Variance Components for Purebred and Crossbreed Data				
	Var(a)	Var(p)	Var(l)	Var(e)	Var(ph)
A	0.912 + 0.123	0.601 + 0.090	0.090 + 0.050	7.787 + 0.070	9.393
B	/	/	/	/	/
C	0.955 + 0.198	1.107 + 0.199	0.120 + 0.126	10.219 + 0.160	12.403
ABC	0.970 + 0.089	0.727 + 0.075	0.181 + 0.039	8.754 + 0.058	10.634
	**h^2^**	**p^2^**	**l^2^**	**e^2^**	
A	0.097 + 0.012	0.064 + 0.009	0.009 + 0.005	0.829 + 0.007	
B	/	/	/	/	
C	0.077 + 0.015	0.089 + 0.016	0.009 + 0.010	0.823 + 0.011	
ABC	0.091 + 0.008	0.068 + 0.007	0.017 + 0.003	0.823 + 0.005	

Var(a)—direct additive genetic variance; Var(l)—variance of common litter environmental effect; Var(p)—permanent environmental variance; Var(e)—residual error variance; Var(ph)—phenotypic variance; **h^2^**—direct heritability; **p^2^**—proportion of permanent environmental effect; **l^2^**—proportion of common litter environmental effect; **e^2^**—proportion of residual error variance.

**Table 4 animals-12-02688-t004:** Connectedness rating (CR) between herds for Large White and Landrace.

Dataset	Herd	A	B	C
	A	1.000000	0.887295	0.875809
Large White	B	0.875334	1.000000	0.983121
	C	0.864112	0.983272	1.000000
Landrace	A	1.000000	0.274184	0.251712
	B	0.269284	1.000000	0.894574
	C	0.245873	0.895578	1.000000

**Table 5 animals-12-02688-t005:** Connectedness rating (CR) between herds for purebred and purebred and crossbred data.

Dataset	Herd	A	B	C
	A	1.000000	0.869794	0.860125
Purebred	B	0.838694	1.000000	0.987166
	C	0.828842	0.986661	1.000000
	A	1.000000	0.971128	0.994608
Purebred and crossbred	B	0.976018	1.000000	0.980043
	C	0.994738	0.975265	1.000000

Purebred—Large White and Landrace; Purebred and crossbred—Large White, Landrace, Large White × Landrace, Landrace × Large White.

**Table 6 animals-12-02688-t006:** Average reliability of prediction in contemporary groups using purebred data.

Herd	Reliability				
	N	$\overset{-}{x}$	SD	Min	Max
A	1194	0.5198505	0.1327724	0.3415963	0.8014904
B	4887	0.7101317	0.0744571	0.3778204	0.9441333
C	3091	0.8672705	0.0213980	0.7678735	0.9430393
ABC	7481	0.7426409	0.0886325	0.5907634	0.9690157

N—number of sows, $\overset{-}{x}$—mean, SD—standard deviation, Min—Minimum, Max—Maximum.

**Table 7 animals-12-02688-t007:** Average reliability of prediction in contemporary groups using purebred and crossbreed data.

Herd	Reliability				
	N	$\overset{-}{x}$	SD	Min	Max
A	14,376	0.8224012	0.0411112	0.7261741	0.9791662
B	/	/	/	/	/
C	10,880	0.8122522	0.0419156	0.6077485	0.9695336
ABC	23,453	0.8460510	0.0414397	0.7423716	0.9885285

N—number of sows, $\overset{-}{x}$—mean, SD—standard deviation, Min—Minimum, Max—Maximum.

## Data Availability

Data used in this research are available from the corresponding authors on request.

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
