# Peer review of "Case Study on Increasing Breeding Value Estimation Reliability of Reproductive Traits in Serbian Highly Prolific Large White and Landrace Sows"

_animals, 2022, doi:10.3390/ani12192688_

Round 1
Reviewer 1 Report
Reviewer
In their manuscript, the authors investigate the possibility of estimating the breeding value of reproductive traits for NBA piglets in three large farms in Serbia, which are also connected through the AI center in one of them. Considering the fact that they study these three farms over a long period of time, that they have artificial insemination and documented pedigrees and zootechnical data, we can conclude that the use of these methods such as gene flow and connectedness is appropriate.
Suggestions for minor corrections
Line 17
I suggest to change the following:
…..larger amounts of data to larger data set of data
Line 33
Heritability values for NBA is expressed in %?
Line 141
Table 1
I suggest adding another, smaller row of breaks between phenotypes, litter genotype, and pedigree file in the table header. The symbol for the average number should be corrected so that the row above x
Please give me some explanations regarding breeding
I am not so familiar with practical application of different methods for analysis of breeding values, but I am closer to theoretical quantitative population genetics and its application in pig breeding practice. It is very important to have real data and include as little "bad" information as possible. So I have a few thoughts that I would like answers to?
In the past, selection was based on the number of teat pairs, which consequently with increasing the number, changed the number of born piglets. All gilts were culled with six teat pairs from nucleus very early in their life? The question arises do we have a selection for NBA or for number of teat pairs or vice versa?
Often in large farms and also in smaller breeding stations, the data on the number of born piglets, born alive or stillborn piglets are not registered on the "sow list" immediately after birth, but it often that this happens a few days later. Any piglet losses that occurred during this time are not counted and the NBA is purged of all losses? How does this work in farms A, B and C?
What happens to the cross-fostering in the first 24-48 hours? Was it done randomly before documenting the data or was it not present at all?
For artificial insemination, the semen of boars from farm B or the progeny of boars from farm B is used? Usually, the insemination stations do not sell the best boars to other farms and very rarely sell their semen? Did farms A and C have access to the semen of the best boars from farm B with at least equal probability or not?
Thanks for yours answers.
Reviewer 2 Report
ID: animals-1916333
Title: Case Study on Increasing Breeding Value Estimation Reliability of
Reproductive Traits in Serbian Highly Prolific Large White and Landrace Sows
Authors: Nenad Stojiljković et al.
Major problems:
1)Typo Errors
There are many typographic Errors.
Table numbers, for example, which table is real 1, 2 or 3 ( “L 260 and L296” are Table 2. “L278 and 306” are Table 3 and no Table1? )
Clarify to the unit of values, (L33: heritability values are “0.048 to 0.097%”, which means “0.00048to 0.00097” ??
There might be wrong unit of “%” in L36, L320, L464 ?
2) Conclusions
L470: authors wrote “ High values for EBV reliability are associated with low heritability and high across-herd connectedness”
Definition of reliability (L234) shows larger VarA, which leads high heritability, then, it causes high reliability?
Why you could conclude that “High values for EBV reliability are associated with low heritability” (not high heritability) ?
Reviewer 3 Report
The manuscript entitled “Case Study on Increasing Breeding Value Estimation Reliability of Reproductive Traits in Serbian Highly Prolific Large White and Landrace Sows” has added gene flow and connectedness information into genetic evaluation of reproductive traits, and leaded a systematic improvement of reliability of EBV estimation. It is valuable and provides a reference for other pig breeders. Before acceptance, there need some modifications for the manuscript.
1. In Abstract as well as in other sections, 58043 should be 58’043, and also including all numbers over 1’000 in the manuscript.
2. The expression of Mixed Model Methodology (MMM) is not a common expression in academic community. Suggest using Linear Mixed Model (LMM) or Mixed Linear Model (MLM).
3. Please carefully check grammatical errors, such as line 58, line328, miss the mark of full stop.
4. Some references maybe have format problem, and the readers cannot know what type of reference, such as 26, 27 and 28. Please check.
5. There should be an explanation or discussion for the fundamental of cross-breed Connectedness rating (CR).
Reviewer 4 Report
The article is interesting, the collected material is impressive, the calculation methods are right. It is justified to conduct selection for reproduction traits despite their low heritability. Notes are marked in the manuscript.
